# Value-Based Healthcare Delivery: A Scoping Review

**DOI:** 10.3390/ijerph21020134

**Published:** 2024-01-25

**Authors:** Mirian Fernández-Salido, Tamara Alhambra-Borrás, Georgia Casanova, Jorge Garcés-Ferrer

**Affiliations:** 1Instituto de Investigación en Políticas de Bienestar Social (POLIBIENESTAR)—Research Institute on Social Welfare Policy, Universitat de València, 46022 Valencia, Spain; tamara.alhambra@uv.es (T.A.B.); jordi.garces@uv.es (J.G.F.); 2Centre for Socio-Economic Research on Ageing, IRCCS-INRCA National Institute of Health & Science on Ageing, 60124 Ancona, Italy; g.casanova@inrca.it

**Keywords:** health systems, healthcare, value-based healthcare, integrated care, older patients

## Abstract

Healthcare systems are transforming from the traditional volume-based model of healthcare to a value-based model of healthcare. Value generation in healthcare is about emphasising the health outcomes achieved by patients and organisations while maintaining an optimal relationship with costs. This scoping review aimed to identify the key elements and outcomes of implementing value-based healthcare (VBHC). The review process included studies published from 2013 to 2023 in four different databases (SpringerLink, PubMed, ProQuest and Scopus). Of the 2801 articles retrieved from the searches, 12 met the study’s inclusion criteria. A total of 11 studies referred to value as the relationship between the outcomes achieved by patients and the costs of achieving those outcomes. Most of the studies highlighted the presence of leadership, the organisation of care into integrated care units, the identification and standardisation of outcome measures that generate value for the patient, and the inclusion of the patient perspective as the most prominent key elements for optimal VBHC implementation. Furthermore, some benefits were identified from VBHC implementation, which could shed light for future implementation actions. Therefore, the VBHC model is a promising approach that may contribute to an improvement in the efficiency and sustainability of healthcare.

## 1. Introduction and Background

Population ageing characterises one of the most important transformations in developed and emerging countries ever since the middle of the last century [1,2]. Along with a progressive increase in older people (65 years of age and older), the development of chronic conditions (multimorbidity), which increase with increasing age [3,4,5], has become a challenge for the provision and financing of healthcare and social services. In this regard, health systems aim to address one of the main concerns about citizen care: the effectiveness of healthcare outcomes [6]. In this field, effectiveness is a term that addresses both the quality of care and the optimisation of care processes [7,8]. Healthcare organisations are currently facing increased pressure on their total expenditure, the increased complexity of people’s health status, and the increased democratisation of therapeutic interventions [9].

In this respect, several studies on the historical development of healthcare have recorded the various changes in the doctor–patient relationship and in the healthcare model [10,11]. In the traditional medical model, the patient is reduced in his or her relationship with the health system and health professionals to a passive subject and a generic person, i.e., without history or context. This model of healthcare began to be redefined in the 1970s [12,13,14]. Since then, various proposals on the therapeutic relationship, guided by the principles of quality, safety, and symmetry, have promoted an increasingly human, ethical, and social interpretation of the patient [15]. The application of these propositional visions has resulted in care provision models moving from ‘patient-centred care’ [16] to ‘person-centred care’ [17]. Moreover, the services of today’s healthcare organisations take into account, as a reference point in the provision of care, both costs and satisfaction, as well as the active participation and experience of the population to be cared for [18,19,20], leading to a new ‘value-based model of healthcare’.

This new paradigm of Value-Based Healthcare (VBHC) is presented as the optimal alternative to the current care approach to health services, the volume-based healthcare model [21]. The proposal of the VBHC model responds to the need to address the costs of health services in relation to their capacity to improve the situation of patients [22]. This need is one of the main challenges facing healthcare organisations today, given the limitation of public resources and the growing complexity, diversity, and plurality of the health status of populations [23,24]. The value-based model of healthcare addresses these material, socio-demographic, and care challenges or constraints of contemporary health provision with a holistic approach to the quality of health services [25]. In this sense, value in healthcare is understood as the trade-off between outcomes and costs, by extension, as the potential effectiveness of health services [26,27,28].

The volume-based model of healthcare operates with a quantitative approach to health service provision. Thus, at the level of clinical performance, the capacity of consultations is prioritised over the patients themselves, and the cost of provision is prioritised over the quality of services [29,30]. As a result, healthcare organisations are delivering healthcare that is increasingly efficient but more segmented by department and with less capacity for improvement [31]. Faced with this clinical trend of the loss of person-centredness, the value-based healthcare model is presented as a strategy that revitalises the active role of the patient and the viability of health services. This new paradigm of healthcare complements health economics with a qualitative and holistic approach to its administration and provision to the population. Thus, it is proposed as a normative healthcare strategy focused on the construction of the value chain of the care process [25]. Even though different initiatives on VBHC have been implemented and analysed over the last few years, there remains a lack of acceptance of the concept and a knowledge gap around the existence of a consensus on the definition of the VBHC concept. This is due to a number of factors, including variations among different health systems around the world and the paucity of available data demonstrating the effectiveness of measures implemented under the VBHC model [32]. There are different interpretations of value and of the key elements for its successful implementation, as well as a multitude of initiatives advocating different positive outcomes. Thus, this study strives to reduce this knowledge gap by bringing together the relevant literature and hopefully laying the groundwork for future research in this area.

Our study aims to explore and synthesise the existing knowledge, through a scoping literature review, of the VBHC conceptualisation and the key elements and outcomes of implementing value-based care in the healthcare context and to identify how these may contribute to improving the efficiency and sustainability of the healthcare system. Therefore, the aim of this study is to identify, compare, and summarise the findings of the literature on the following: (1) the definitions of value-based care extracted from the literature review; (2) the key elements of implementing/delivering value-based care into the healthcare context; and (3) the main outcomes, in terms of improvement in the quality of the care process, of implementing value-based care. Moreover, this scoping review aims to explore and identify pertinent gaps that would be beneficial for guiding future studies.

## 2. Materials and Methods

A scoping review responds to a specific methodology of locating and selecting existing studies on a specific topic, according to pre-specified eligibility criteria, in order to analyse, synthesise, and report the results found, with the aim of answering a clearly specified research question [33]. We adopted the guidelines of the PRISMA 2020 statement (Preferred Reporting Items for Systematic Reviews and Meta-Analyses) for its relevance to this study, as it is primarily designed to evaluate systematic reviews of studies assessing the effects of health interventions, regardless of the design of the studies included [34,35]. In particular, the PRISMA extension for Scoping Reviews (PRISMA-ScR) checklist was used to guide the conduct of this review; the protocol of this scoping review has not been registered or published [36].

The bibliographic reference management application Zotero was used to transfer the studies identified in the electronic databases and eliminate duplicates. To maximise the evidence recovery from the databases, the literature search strategy was based on the PICO framework, consisting of the construction of the research question according to the description of the following components: patient or problem of interest (P), intervention (I), comparison (C) and outcomes (O) [37]. However, this study does not present an intervention with which to compare value-based care, so component C is omitted, leaving the format as PIO. Table 1 shows the PIO model.

After using the PIO model, the following research questions are presented:

Does the implementation of a value-based approach lead to an improvement in the efficiency and sustainability of healthcare? 

If so, which are the key elements and the related outcomes of implementing the value-based care approach in the healthcare context?

In order to identify, select, and include relevant literature that answered the research question, and to discard studies that did not answer it, inclusion and exclusion criteria were defined. The following inclusion and exclusion criteria were used for study selection:

### 2.1. Eligibility Criteria

Inclusion criteria:(1)Articles addressing the implementation of value-based care in the healthcare context;(2)Articles published in the last 10 years (2013–2023);(3)Articles published in English and Spanish;(4)Articles published in scientific journals;(5)Full and open access articles;(6)Original or primary source studies that are descriptive, experimental, quasi-experimental, cross-sectional, and longitudinal articles.

Exclusion criteria:(1)Articles that do not address the implementation of value-based care in the healthcare context or articles focused on a specific condition/disease;(2)Articles that were published more than 10 years ago;(3)Studies that were published in a language other than English and Spanish;(4)Articles published in non-scientific journals or incomplete and non-open access articles;(5)Secondary source studies, such as reviews and editorials.

### 2.2. Search Strategy

A scoping literature review was conducted according to the inclusion and exclusion criteria above in the electronic databases PubMed, ProQuest, Scopus, and SpringerLink. The databases were chosen for their international recognition and broad multidisciplinary coverage, with the intention of retrieving articles relevant to the subject of the scoping literature review. According to the eligibility criteria, articles that included descriptors related to the terms MeSH (Medical Subject Headings) and DeCS (Health sciences desCriptors) were selected [38]. The terms value-based, care, health, and healthcare were identified from the MeSH and DeCS descriptors and combined with a Boolean operator, as shown in Table 2, to develop a database search to achieve the proposed objectives.

To reduce the risk of subjective interpretation and possible inaccuracies due to chance errors that might have affected the results of the review, two independent reviewers were involved in the selection of studies in the electronic searches [39]. Thus, after eliminating duplicate records, we proceeded with the preliminary data analysis, which included a three-stage selection process: the first phase based on selection by title, the second phase based on selection by abstract, and the third phase consisting of reading the full text of the studies selected by abstract. Each of the papers was assessed twice by two independent reviewers following the inclusion and exclusion criteria set above. A third reviewer was involved in the process when disagreement arose or consensus was not reached, making the final decision.

## 3. Results

### 3.1. Screening Results

A total of 2.801 records were identified covering the time span of January 2013 to December 2023, of which 1.609 duplicate registrations were deleted. Of the 1073 records obtained after eliminating duplicates, 439 records were identified in the title review. After applying the exclusion criteria in the selection by abstract, 396 were eliminated, leaving 43 scientific articles for full-text review. A total of 12 full text articles were reviewed, all of which met the inclusion criteria and were included in the final list of studies included for this research. To conduct and report this scoping review, we used the preferred reporting elements for systematic reviews and meta-analysis scoping reviews: PRISMA-ScR [34] (Appendix A) together with the PRISMA 2020 flowchart [35] (Figure 1). Figure 1 presents the flow diagram, which was based on the PRISMA 2020 declaration [35], that illustrates the scoping literature review process and details the reasons for exclusion at each screening stage.

#### Preliminary Insights

A total of 12 studies were included for the scoping review. In terms of the methodology used in the studies, 10 of the 12 studies used qualitative techniques, either through interviews or focus groups, and 2 other studies used a mixed methodology, combining the use of interviews and questionnaires. The following table (Table 3) shows the PICOTS characteristics for each of the studies included in the scoping review [40]. Table 4 shows a summary of the results found from the analysis of the studies in the scoping review.

### 3.2. Results and Analysis

#### 3.2.1. Definitions of Value-Based Care

The first specific objective was to identify and compare the definitions of value-based care extracted from the scoping literature review. All the definitions found in the reviewed literature placed the patient at the centre of the definition of VBHC. Nilsson et al. [41,42] described VBHC as an approach based on three principles: first, creating as much value as possible for the patient; second, basing the organisation of healthcare on the patients’ medical conditions and full care cycles; and third, the measurement of medical outcomes and costs.

Aligned with the first principle highlighted by Nilsson et al. [41,43]; Steinman et al. [48,50] stated that ‘value consists of what matters most to patients’, while Daniels et al. [44] emphasised that patient value is defined as ‘the best possible patient-relevant health outcomes and patient experience divided by the costs to achieve those outcomes’. In this line of thought, Heijster et al. [45] explained that a key element of VBHC is ‘to improve outcomes in daily practice that matter to patients while optimizing resource utilization’.

The optimisation of resource utilisation has been also mentioned in the definitions found in Cossio Gil et al. [32]; Makdisse et al. [46]; Ng, S. [47], and Steinman et al. [48]; in these, it was mentioned that, within a VBHC approach, improving value requires improving outcomes per unit of cost. Thus, the importance of measuring both health outcomes and costs, as supported by Nilsson et al. [41,42,43] in their third principle of VBHC, is highlighted. This idea was also supported by the definition extracted from a study by Makdisse et al. [46], according to which ‘the value equation is where value is defined as health outcomes relative to the cost’.

Furthermore, for Cossio Gil et al. [32], VBHC must put patient outcomes at the centre of the healthcare process. This idea is in line with the second principle of VBHC from Nilsson et al. [39,41], regarding basing the organisation of healthcare on the patients’ medical conditions. This has also been supported by the work of Makdisse et al. [46], who recommended a value agenda in which healthcare should be organised into integrated practice units. This authors, as part of this value agenda, also mentioned that healthcare systems should move to bundled payments for care cycles and that information technology platforms must be enabled in order to achieve VBHC.

Cossio Gil et al. [32] also stated the importance of VBHC for professionals, as it can be a key aspect for reducing the burden on professionals and improving satisfaction with their work.

Finally, Verela-Rodríguez et al. [49] described VBHC as ‘an international trend that implies significant changes at several levels of the healthcare institutions from managerial viewpoints to the doctor–patient relationship’.

The analysis of all the selected studies confirms the presence of commonalities among the identified definitions. For instance, the term ‘patient’ is consistently present in all definitions, and the correlation between outcomes and costs is observed in 72.7% of the studies. Yet, the distinctions among the definitions emerge in terms of how they articulate the correlation between costs and outcomes. This connection is occasionally directly associated with the VBHC concept, while in other cases, it is associated solely with the definition of ‘value’ as a constituent within the concept. Additionally, references to supplementary factors beyond patient outcomes and costs, like the impact on professionals’ workload and their job contentment, underscore the presence of a gap in comprehending the concept.

#### 3.2.2. Key Elements of Implementing and Delivering Value-Based Care

The second specific objective of this present scoping literature review was to identify the key elements of implementing and delivering value-based care into the healthcare context.

Among the reviewed studies, the following key elements have been identified: leadership, involving the patients’ perspective, organising the delivery of care in integrated care units, the standardisation of outcome measures and accessibility of data, and having enough resources in terms of time and human capital.

A total of nine studies considered the presence of leadership as a key element to support and guide the (multidisciplinary) teams implementing the VBHC approach within the hospital: Nilsson et al. [41,43]; Hejister et al. [45]; Daniels et al. [44]; Cossio et al. [32]; NG [47]; Steinman et al. [48] and Varela et al. [49]. According to Nilsson et al. [41,43], effective leadership occupies a role within the team that is persevering, committed throughout the process, able to motivate and drive the team, and is constantly able to bring new ideas and approaches. This effective leadership was considered essential to ensure that the implementation does not slow down or even that the value-based work does not come to an end.

Hejister et al. [45]; Daniels et al. [44], and Cossio et al. [32] highlighted that effective leadership is based on ensuring the involvement of patients and/or patient representatives, as well as the necessary financial resources for the successful implementation of VBHC. Likewise, Hejister et al. [45] highlights the figure of the clinical leader, and Daniels et al. [44] highlights the figure of the medical leader as figures responsible for leadership in order to successfully launch the implementation of the model. While for NG [47], in the frame of VBHC, great leaders are those that support the implementation of changes and reforms to ensure organisational efficiency with clear pathways for patients [47].

On the one hand, several of these studies focused on the importance of leadership in structuring the work among the team in the pre-implementation phases of VBHC [43,48]. In this sense, studies confirm that leadership by the hospital director, according to which the VBHC approach should be used as a management tool, allows for the legitimacy of decisions within the teams and is conceived as crucial for the prior organisational redesign necessary for the subsequent successful implementation of VBHC [43,48]. On the other hand, another study highlighted the relevance of leadership in both the pre-implementation phase and also in the leading of the implementation process to ensure the motivation of the team during the first months [41]. Although, without providing details, other studies also allude to leadership and coordination as a key step in ensuring the successful implementation of VBHC [49].

Studies also agree on the importance of involving the patients’ perspective, although they differ in their manner. Some of the studies emphasised that the patient is at the core or centre of VBHC [43,47,51]. In the same line, other studies highlighted the importance of involving patients or patient representatives during the implementation process [41,45]. According to Nilsson et al. [41], patient involvement is key to understanding the patients’ point of view and to ensure that there are no discrepancies between patients’ experiences of value and how teams implement VBHC. In this sense, involving patients or patient representatives allows teams to seriously evaluate care delivery in relation to patient value [41]. In the same vein, other studies highlight that patients as well as teams need to have access to data in order to discuss changes in the care process together [32]. Finally, other studies confirmed that VBHC contributed to highlighting the importance of including the patients’ perspective and what is important to them [43].

Other studies emphasised that patients’ involvement alongside the multidisciplinary team needs to be present not only at the implementation phase but also during the pre-implementation design process. In this sense, patients are considered members of the value team, and their participation is essential to ensure personalised care in which their wishes and needs are included, and the outcomes that will be relevant to measure in later stages are selected [45]. Other studies considered the patient perspective to be essential when implementing VBHC, because patients’ perspective is key to developing tools that are relevant to actually assess patient-reported outcomes (PROMs) and patient experience (PREMs) through systematic measurements [32,49].

Other studies also mentioned involving patients in the shared decision-making process as one of the most important elements of VBHC [32,50,51].

Another key element for VBHC is embedded according to the studies in the pre-implementation phase, known by some studies as organisational [48], or more generically, they refer to the organisational structure of hospitals [45,47].

In this respect, the studies emphasised that, prior to the implementation of VBHC, it is essential to modify the healthcare organisation, which is usually organised in separate departments, into integrated care units [32,41,44,49,50]. According to these studies, healthcare systems that are organised in specialised departments make it difficult to assess outcomes, to measure costs along the whole process, and to follow patients during the course of the disease as they move from one department to another. For these reasons, it is considered necessary to organise care delivery in integrated care units or in multidisciplinary care pathways around a specific patient group with a specific medical condition [32,40,48,49,50] or, in other words, towards a disease-oriented organisation that allows the entire care process to be evaluated in terms of costs and clinical outcomes [48].

Standardisation of outcome measures and accessibility of data: Importance of ICTs.

As previously said, patients’ involvement is essential to know what value for patients is. Thus, the identification of outcome measures relevant to patient groups, which creates value for patients, is another key element in the implementation of VBHC [32,41,47,48,49]. Alongside the identification of outcome measures, the studies highlight the importance of new technologies for recording and accessing outcomes which facilitates the implementation of VBHC. Several examples that confirm that IT support is an important factor for a successful delivery of VBHC are presented in the reviewed literature. These include the following: the creation of information platforms that enable communication and inform both clinical teams about PROMs and patients about their health status [32], the development of a coding system to measure outcomes across a whole group of patients [42], the installation of supporting IT tools that allow for the searching of data in different IT systems of a hospital [42,51] or that allow, in a given hospital, the systematic recording of information from the primary source, the existence of an up-to-date IT system containing the data, the opportunity to search for statistics for outcome measurement mapping [41], or even the presence of national data registers [44].

Alongside the measurement of outcomes, several studies highlight the importance of measuring the costs of the entire care cycle [40,48,49]. Along these lines, some studies highlight that, in order to calculate the value for patients, it is necessary to measure the costs per patient of the entire care process [49,50] or, in other words, to measure the costs of the care cycles for each of the diseases they treat [48].

A few studies also highlighted the importance of having enough resources available during the design and/or implementation of VBHC for the successful implementation of this approach. In this regard, time was considered one of the most important resources in many studies [41,43,44].

When planning VBHC implementation, time was found to be essential in order to ensure the sufficient preparation of the teams to understand the meaning of VBHC and what value-based work implies, to decide on the administrative resources needed for the implementation process [43], to adjust the essential IT systems that would be key during the implementation [41,43], and to detect, with the staff involved in the teams, which results were interesting to measure the amount of time necessary to schedule the required follow-up meetings to monitor the implementation process [50]. Once VBHC was implemented, time was seen as a key resource to reflect and adapt to all changes without losing track of the work being done [40,43,44]. Apart from time, human capital was also found as a key resource for the successful implementation of VBHC. Several studies highlighted the importance of having multidisciplinary teams for VBHC implementation to ensure integrated and multidisciplinary value-based care [44,45,49,50]. Other studies highlighted the hiring of external consultants as a key figure to structure the work in the pre-implementation phase and to lead the implementation process in the first months, supporting, guiding, and motivating the teams during the implementation process [41,45]. In other cases, it was also considered essential to involve professionals outside the VBHC team, as it was felt that all professionals involved in the care of a given patient group should support improvement initiatives [44].

Despite the recent implementation of this model, there is consistency across studies regarding the critical elements necessary to guarantee its effectiveness in implementation. In 75% of the studies, both the leadership and the organisation of integrated care units emerge as prominent elements. Moreover, the significance of involving patients, ensuring data accessibility, and updating IT systems is evident in over half of the studies. These examples signify a substantial consensus among stakeholders regarding the essential components aimed at enhancing care within a value-based care model.

Nevertheless, despite these commonalities, the differences among studies, even those addressing identical elements, are striking. Certain studies mention leadership yet diverge in attributing this role to various individuals, such as the hospital director, or remain vague about who should possess this capacity. Moreover, there are differing perspectives on its significance at different stages, with some emphasising its importance during the pre-implementation organisational phase, while others highlight its relevance specifically during the implementation of value-based care. Similar variability exists regarding the inclusion of patients in these studies. While some emphasise considering the patients directly, others discuss the potential inclusion of patient representatives. The discrepancies in identifying crucial elements for the effective adoption of the value-based care model imply a lack of uniformity in the understanding of its implementation. Consequently, this variation in understanding could lead to differences in measurement and outcomes, thereby complicating comparisons between implementations.

#### 3.2.3. Resulting Outcomes of Implementing Value-Based Care

The third specific objective of this scoping literature review was to identify and summarise the resulting outcomes, in terms of improvement in the quality of the care process, of implementing VBHC, and how this may contribute to improving the efficiency and sustainability of a healthcare system. Studies show positive results from the implementation of VBHC, including improvements in the awareness of cooperation and actual cooperation between the different departments involved in the patient care process and a better understanding of the different conditions in each department and different patient subgroups. This improved coordination, together with VBHC, facilitated the achievement of outcome measurements and improved the quality of data. In terms of patients, the implementation of VBHC increased accessibility for patients to receive care at the appropriate level of care and better patient follow-up. In general, VBHC implementation increased the sustainability of organisations, in particular of those where professionals were committed, and strong leadership was in place.

Different studies referred to certain human resources involved in VBHC implementation for their ability to guide positive outcomes [44,48]. In this sense, the studies highlighted that involving professionals from outside the VBHC team (e.g., from other hospitals) who care for a certain group of patients allows for the professionals to share the actions that are being implemented with them and increase knowledge about the best practices [50]. The specific presence of other professional profiles, such as managers, also was mentioned due to their ability to foster communication between the different care units involved in the full care cycle of a disease [48].

Generally, some studies highlight that working according to a standardised care plan contributes not only to a greater awareness on the part of professionals to use hospital time efficiently and a better structured care process but also to the higher job satisfaction of the staff [42].

Other studies have highlighted that, in the implementation of VBHC, the commitment and input of physicians and managers, together with clinical leadership, enabled organisations to innovate and drive changes and reforms, achieving greater efficiency in hospital services [44,47,48]. In this sense, some studies highlight that the most successful and sustainable organisations have been those in which there was a greater degree of commitment between doctors and managers [47]. Along the same lines, other studies have highlighted physician leadership as a success factor due to the positive involvement of the physician leader as an inspirational and motivating character with the ability to involve others and assume responsibility [44], which led to a successful delivery of VBHC. Finally, some studies highlight leadership within implementation teams as very beneficial for the proper organisation of teams [48].

On the other hand, the studies highlighted that emphasising value for patients brings benefits for the healthcare organisation implementing VBHC, as it enables (team) participants to understand the patients’ point of view, become enthusiastic about the concept, and strongly engage in implementation work [41].

Another important outcome of VBHC implementation was organisational improvement in terms of increased cooperation between departments and between professionals in these departments. In turn, this improved cooperation facilitated the achievement of outcome measurements, patient follow-up, and the understanding of the different conditions in each department and different patient subgroups [41]. In terms of improving cooperation on a broader level, the implementation of VBHC also increased the awareness of cooperation between inpatient and outpatient care, contributing to increased accessibility for patients to receive care at the appropriate level of care [41,42].

The creation of integrated units around medical conditions also triggered positive consequences by considering the fact that they could enable closer collaboration between all those involved in the treatment of patients with a particular medical condition and allow hospitals to better address the interdependencies of the different activities necessary for patient care [43].

Finally, studies highlight that the implementation of VBHC improved data quality by using systematic measures to actually assess patient-reported outcomes (PROMs) and patient experience (PREMs) as well as enter the information into the system from the primary source (physician/patient). This reduced interpretation bias, ensured systematic recording, and avoided missing data [46,48]. More briefly, other studies conclude that the use of patient-reported outcome measures has itself been a stimulating factor for the implementation of VBHC [44]. Furthermore, the transparent display of health outcome information, so that it is available to both care providers and the general public, has also been shown to facilitate improvements in the health outcomes achieved [51]. In the same vein, other studies confirm that having a coding system to measure health outcomes in a subgroup of patients allows the team to critically examine processes and decisions in relation to different treatment regimens [42]. More generally, other studies have emphasised that value-based metrics have a driving effect on collaboration among team members by creating a sense of shared accountability for certain goals [48].

In conclusion, the studies included in this scoping review present results (66.6%) that refer to predominantly positive outcomes. These studies correlate these favourable outcomes with the presence of key elements highlighted in the implementation of VBHC. Nevertheless, a notable proportion of studies (33.4%) within this review do not present specific outcomes or results. Furthermore, the disparities observed in the examined results are due to the absence of a standardised foundation for the selection of key elements and their implementation.

## 4. Discussion

This review describes the state of the art regarding the concept of VBHC, key elements for its successful implementation, and the resulting positive outcomes of implementing VBHC within a healthcare system.

In terms of the VBHC conceptualisation, the definitions found in this literature review referred to both the general term VBHC and the meaning of value within the model. Most of the studies agree on the definition of value and define it as the health outcomes achieved for patients in relation to the costs of the whole process of care [27,35,52]. In this sense, delivering value to the patient means improving health outcomes for the patient. This definition of value is aligned with the definition of value of Michael Porter and Elizabeth Teisberg in their 2006 book on redefining healthcare [26], with these authors in this particular work being the pioneering authors of the VBHC approach.

Despite the unanimity in the definition of value, studies vary in their consideration of the key elements or factors in the implementation of VBHC. This ambiguity in the conception of the term has resulted in multiple ways of implementing VBHC depending on the geographical context and management of health systems [53]. This study may contribute to unveiling this cloak of ambiguity about the key elements of VBHC implementation presented in the scientific literature.

Thus, with regard to the key elements of VBHC, those most frequently examined were, firstly, the existence of a leader with the capacity to motivate and guide the team in the pre-implementation and implementation phase; secondly, the involvement of patient perspectives to ensure that the implementation of VBHC is responsive to the patient experience and to guarantee personalised care; thirdly, the creation of integrated care units around specific patient groups or specific medical conditions that allow patients to be followed throughout the process; fourthly, the identification and storing of patient perspectives to ensure that the VBHC implementation responds to patient experiences and guarantees personalised care; fifthly, the identification and standardisation of relevant outcome measures for patients in conjunction with the development or improvement of IT systems to ensure the recording, transparency, and accessibility of data by care providers and patients; and finally, the provision of time and human resources to ensure that implementation teams have the necessary time for preparation and the necessary reforms prior to implementation and for monitoring and adapting to changes during the implementation process. These elements have been identified in a wide variety of scientific studies [45,49], and it is recognised that their combination is considered essential for VBHC implementation. The pioneering work of Porter and Teisberg [26], as well as their further research, has shown that the transformation from volume-based care to value-based care must be based on a combination of six elements: organising around integrated care units, measuring outcomes and costs per patient, bundled payments by care cycles, expanding geographic reach, and enabling an informatics platform, with most of them being aligned with the key elements of implementing VBHC found in this scoping review.

Regarding the identification of positive outcomes resulting from the VBHC implementation, some benefits have been identified that could shed light for future implementation actions.

Among them, some of the reviewed studies described improvements in cooperation between professionals working in the healthcare system, both in terms of raising awareness of the need of cooperation and improvements in actual cooperation between professionals and departments involved in the patient care process. Cooperation has been shown to be essential for optimal care provision in other studies [54,55]. In addition, it was described in several of the reviewed studies [41,42] that the creation of integrated units was also seen as beneficial in enabling closer collaboration between all those involved in the treatment of patients with a disease and between the different levels of care (inpatient and outpatient). This improvement is supported by the ‘integrated care’ approach that seeks to better coordinate care around people’s needs [56]. Along these lines, it was also found that the implementation of VBHC increased accessibility for patients to receive care at the appropriate level of care and better follow-up. Other positive outcomes of the delivery of VBHC are that implementing this model facilitated the achievement of outcome measurements and the quality of the data collected. As widely highlighted by the ICHOM—International Consortium for Health Outcomes Measurement—group, measuring outcomes is important to deliver optimal healthcare that matters to patients. Thus, the improvement found in our literature review in those terms are aligned with the ICHOM group’s vision, as they contribute to value maximisation, where value is understood as ‘the best possible patient-relevant health outcomes and patient experience divided by the costs to achieve those outcomes’ [44]. Previous studies confirm the high degree of the interpretive variability of the concept, as well as the lack of consensus on its conceptualisation and the paucity of information on the evaluation of the strategies implemented [57]. In this sense, the present scoping review addresses the interpretative variability and differences in the conceptualisation of VBHC, providing an individual and comparative analysis of the studies included, thus adding value to previously published studies that agree on the existence of a gap around a generalised definition and understanding of the model. In addition, this study sought to address the paucity of results reported in previous studies on the evaluation of the implementation strategies in place by providing a comprehensive analysis of the positive results reported in these studies.

Despite the meaningful contributions of this literature review, this study is not without its limitations. First, our study protocol was not prepared neither registered, as recommended by the PRISMA 2020 guidelines. Moreover, our search was limited to studies published in English and Spanish between 2013 and 2023, which may exclude studies published in other languages that might be relevant to understand VBHC. In addition, most of the studies included in this literature review are based on a qualitative methodology, which may limit the extent to which the findings of this study can be generalised, and a number of the reviewed studies simply narrate experiences without assessing the effectiveness of implementing the system-wide intervention, which presents a major limitation, as there are no data to guarantee that these interventions work. We believe that there is sufficient consistency in the results analysed in this scoping literature review to be useful in guiding future research, even though the identified limitations suggest the need for additional research to address the gaps in our understanding of this critical healthcare paradigm, as well as on the scalability and sustainability of the VBHC model.

## 5. Conclusions

In conclusion, based on the findings of this scoping literature review, the implementation of VBHC may contribute to an improvement in the efficiency and sustainability of healthcare.

While most studies refer to some of Porter and Teisberg’s key elements, there is no agreed generalisation of all of them, and there is interpretative variability that translates differently in the way VBHC initiatives are implemented and the variety of positive outcomes achieved in terms of effectiveness and the sustainability of healthcare.

These findings point to an urgent need for a common conceptualisation of VBHC, focusing on key elements to reduce interpretive variability and to achieve a shared understanding of its application.

## Figures and Tables

**Figure 1 ijerph-21-00134-f001:**
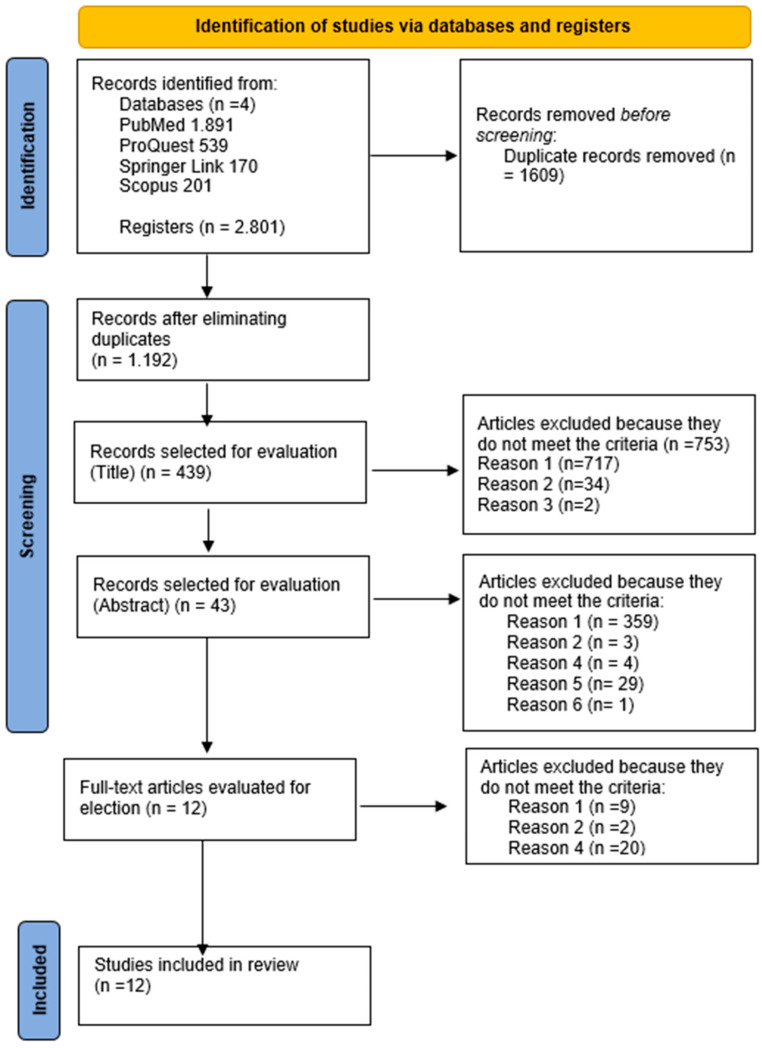
Flow diagram for our scoping review based on PRISMA. Note: Reason 1: Articles that do not address the implementation of value-based care in the healthcare context or articles focused on a specific condition/disease; Reason 2: Articles that were published more than 10 years ago; Reason 3: Studies that were published in a language other than English and Spanish; Reason 4: Articles published in non-scientific journals or incomplete and non-open access articles; Reason 5: Secondary source studies; Resource 6: Duplicate.

**Table 1 ijerph-21-00134-t001:** Elaboration of the research question using the PIO model.

Acronym and Components	Description on Components
(P) Population	Healthcare system at large
(I) Intervention	Value-based care approach
(O) Outcomes	Improving the efficiency and sustainability of healthcare systems, that is to say, the overall outcomes of care in terms of the quality of care.

**Table 2 ijerph-21-00134-t002:** Search with descriptor and Boolean AND operators.

Descriptor	Boolean Operator	Descriptor	Boolean Operator	Boolean Operator	Descriptor
Value-based	AND	Care	AND		
Value-based	AND	Care	AND		
Value-based	AND	Care	AND		
Value-based	AND	Care	AND		
Value-based	AND	Care	AND	AND	Cost
Value-based	AND	Care	AND	AND	Quality of life

**Table 3 ijerph-21-00134-t003:** PICOTS characteristics of reviewed studies.

Reference	Populations	Intervention	Country	Method/Outcomes	Timeframe	Setting
Nilsson et al. [41]	Professionals of VBHC implementation teams in a hospital	Exploration of how the representatives of four pilot project teams experienced implementing VBHC in four different groups of patients	Sweden	Qualitative analysis by conducting interviews over three periods with 20 members of the VBHC implementation teams, with a total of 59 interviews conducted	2 years	A Swedish university hospital
Nilsson et al. [42]	Professionals of VBHC implementation teams in a hospital	Exploration of four pilot teams’ experiences of improvements resulting from the implementation of VBHC in a hospital	Sweden	Qualitative analysis through in-depth interviews with 20 members of the VBHC implementation teams	2 years	A Swedish university hospital
Nilsson et al. [43]	Professionals of VBHC implementation teams in a hospital	Exploration of four pilot teams’ experiences of improvements resulting from the implementation of VBHC in a hospital	Sweden	Qualitative analysis through in-depth interviews with 20 members of the VBHC implementation teams	2 years	A Swedish university hospital
Cossio Gil et al. [32]	Members/professionals of EUHA	Presentation of a plan for the implementation of VBHC in hospitals	Europe	Qualitative analysis	2 years	European University Hospital Alliance (EUHA)
Daniels et al. [44]	Members of VBQI teams in a large Dutch top clinical teaching hospital	Exploration of the main hindering and/or supporting factors in the implementation of VBQI teams in hospital care	The Netherlands	Qualitative analysis with semi-structured interviews with 43 members of 8 VBQI teams	5 years	Dutch top clinical teaching hospital
Heijster et al. [45]	Members of Amsterdam UMC	Presentation of a pragmatic step-by-step approach for VBHC implementation, developed and applied in Amsterdam UMC	The Netherlands	A method for implementing VBHC in Amsterdam UMC based on ‘experience-based co-design’ (EBCD)	3 years	Academic hospital in the Netherlands
Makdisse et al. [46]	Top- and middle-level executives from 70 healthcare provider organisations (HPOs)	Investigation of how HPOs in five Latin American countries were implementing VBHC	Argentina, Brazil, Chile, Colombia, and Mexico	Mixed methods research using online questionnaires and semi-structured interviews with a total of 70 participants from health organisations in five Latin American countries	2 years	Healthcare provider organisations (HPOs) in Latin America
NG [47]	Managers and clinicians of the NHS in the United Kingdom	Exploration of relationships, behaviours, and perceptions between managers and clinicians regarding value-based healthcare	United Kingdom	A qualitative research methodology of semi-structured in-depth interviews applied to a sample of 4 hospital consultants, 4 senior managers, and 4 board executives	---	The National Health Service in the United Kingdom
Steinman et al. [48]	Representatives of Dutch hospitals (the Netherlands)	Exploration of the ways in which Dutch hospitals were implementing and pursuing value-based redesign	The Netherlands	Qualitative study through semi-structured interviews and focus groups with representatives of Dutch hospitals	---	Hospital organisations in the Netherlands
Verela-Rodríguez et al. [49]	Professionals and Members of the Population attending the Hospital Universitario 12 de Octubre	Value-based healthcare project implementation in a hierarchical tertiary hospital	Spain	Pilot study for the implementation of VBHC, in which qualitative techniques such as focus groups and the Delphi technique were included	4 years	Hospital Universitario 12 de Octubre Madrid
Steinman et al. [50]	A Dutch expert panel about VBHC consisting of nine members	The generating of a consensus on key actions and practices for VBHC implementation	The Netherlands	Qualitative research using the Delphi technique with a group of 9 Dutch experts on actions and practices that would contribute to implementing VBHC in the Dutch healthcare system	---	The healthcare system in the Netherlands
Krebs et al. [51]	Members of Germany’s healthcare system	Exploration of stakeholders’ perspectives on the relevance and feasibility of actions and practices related to the implementation of VBHC in the German healthcare system.	Germany	Mixed methodology through interviews and questionnaires (using the Delphi method) with health experts	2 years	The healthcare system in Germany

**Table 4 ijerph-21-00134-t004:** List of studies included for the scoping review.

Reference	Methods	VBHC Definition	Key Elements of Implementing VBHC	Outcomes of VBHC Implementation
Nilsson et al. [41]	Qualitative	Value is defined as health outcomes achieved per “dollar” spent. VBHC implies creating value for patients; basing the organisation of medical practice on medical conditions and care cycles; and measuring medical outcomes and costs.	- Organising healthcare around integrated care units; involving patients or patient representatives; the identification of outcome measures that create value for patients; the accessibility of data—up-to-date IT systems; time; the presence of leadership; measuring the costs of the entire care process.	- Patients appreciated the value-focused care they received; practitioners were more aware of what creates value for patients; increased co-operation between departments and the professionals working in them. - Increased awareness about the necessity of cooperation between inpatient and outpatient care; increased accessibility to patients by receiving care at the appropriate level; improvements in outcome measurement, patient follow-up, and the understanding of different conditions in each department and different patient subgroups.
Nilsson et al. [42]	Qualitative	Patient value refers to the quality of care and treatment provided.	- The presence of leadership; the identification of outcome measures that create value for patients; the accessibility of data—up-to-date IT systems; involving patients.	- Better quality of care; increased staff job satisfaction. - Improved communication between inpatient and outpatient care; improvements in outcome measurement and patient monitoring.
Nilsson et al. [43]	Qualitative	Value is defined as health outcomes achieved per ‘dollar’ spent. VBHC implies creating value for patients; basing the organisation of medical practice on medical conditions and care cycles; and measuring medical outcomes and costs.	- The presence of leadership; time; planning and preparation in the pre-implementation phase.	- It enhanced the importance of including the patients’ perspective and what is important to them.
Cossio Gil et al. [32]	Qualitative	VBHC means improving outcomes for patients in relation to the costs of care while reducing the burden on professionals and improving their job satisfaction.	- Organising healthcare around integrated care units; the accessibility of data—up-to-date IT systems; the identification of outcome measures that create value for patients; involving patients; the presence of leadership.	---
Daniels et al. [44]	Qualitative	VBHC is defined as the best outcomes for the patient divided by the costs of achieving those outcomes.	- Organising healthcare around integrated care units; the presence of leadership; accessibility of data—up-to-date IT systems; time; organisational readiness in the pre-implementation phase; involving professionals from outside the VBHC team.	- Increased knowledge of best VBHC practices. - Increased efficiency in hospital systems.
Heijster et al. [45]	Qualitative	VBHC is defined as the improvement of patient outcomes in relation to the optimal use of resources.	- Organising healthcare around integrated care units; the presence of leadership; involving patients or patient representatives; organisational readiness in the pre-implementation phase; multidisciplinary VBHC implementation teams; the hiring of external consultants.	- Improvement in the process of care by ensuring the inclusion of patients’ wishes and needs; the use of systematic measures to assess patient outcomes reduced interpretation bias, ensured consistent recording, and avoided missing data.
Makdisse et al. [46]	Quantitative and qualitative	Value is defined as the ratio of health outcomes to costs for each patient.	- Organising healthcare around integrated care units.	---
NG [47]	Qualitative	Value requires improved results per unit cost.	- Organising healthcare around integrated care units; organisational readiness in the pre-implementation phase; the presence of leadership; the accessibility of data—up-to-date IT systems.	- Increased efficiency in hospital systems.
Steinman et al. [48]	Qualitative	At VBHC, value is what matters most to patients. Value is defined as the health status of the patient (outcomes) divided by the resources required to achieve that status (costs).	- Organising healthcare around integrated care units. - The presence of leadership; Multidisciplinary VBHC implementation teams; the identification of outcome measures that create value for patients; measuring the costs of the entire care process.	- Improved communication between the different care units involved in the complete care cycle of a disease; improved efficiency of hospital services. - Improved collaboration between team members by creating a sense of shared responsibility for certain objectives.
Verela-Rodríguez et al. [49]	Qualitative	VBHC is defined as an international trend that involves significant changes at various levels of healthcare institutions, from management to the doctor–patient relationship.	- Organising healthcare around integrated care units; the presence of leadership; the accessibility of data—up-to-date IT systems; involving patients or patient representatives; the identification of outcome measures that create value for patients; measuring the costs of the entire care process.	- Reduction in interpretation bias and improvement of data quality (thanks to PROMs).
Steinman et al. [50]	Qualitative	Value is defined as patient health status (outcomes) divided by the resources needed to achieve it (costs).	- Organising healthcare around integrated care units; involving patients or patient representatives; the identification of outcome measures that create value for patients.	---
Krebs et al. [51]	Quantitative and qualitative		- Involving patients or patient representatives; the accessibility of data; up-to-date IT systems; multidisciplinary VBHC implementation teams.	---

Note: Several studies that were examined did not provide information on certain aspects being reviewed, which explains the absence of data in specific table cells.

## Data Availability

No new data were created or analysed in this study. Data sharing is not applicable to this article.

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
