# Peer review of "Value-Based Healthcare Delivery: A Scoping Review"

_ijerph, 2024, doi:10.3390/ijerph21020134_

Round 1
Reviewer 1 Report
Comments and Suggestions for Authors
The methodology used is not appropriate for such a comprehensive topic, especially since research with similar results has already been published!
Search better already published research.
Narrow the focus of research on a specific group of health problems or interventions (VBHC is too broad a topic and general research in this area has already been published).
Pay attention to overlaps with other works. For example
van Staalduinen, D.J., van den Bekerom, P., Groeneveld, S. et al. The implementation of value-based healthcare: a scoping review. BMC Health Serv Res 22, 270 (2022). https://doi.org/10.1186/s12913-022-07489-2
This study showed that VBHC has a high level of interpretative variability….
While most hospitals stick close to the ideas of Porter & Teisberg …
A common conceptualization of VBHC is urgently needed….
Author Response
Comment 1: The methodology used is not appropriate for such a comprehensive topic, especially since research with similar results has already been published!
Response: We really appreciate your comment. The main reason to conduct a Scoping Review instead of a Systematic Review is that actually our paper is not aimed to produce a critically appraised and synthesized result/answer to a particular question, but it is rather aimed to provide an overview or map of the evidence on value-based care approach.
Comment 2: Search better already published research.
Response: Thank you for your concern. We have exhaustively followed a Scoping Review process following PRISMA guidelines.
Comment 3: Narrow the focus of research on a specific group of health problems or interventions (VBHC is too broad a topic and general research in this area has already been published).
Response: We thank you for your comment and fully understand what you are referring to, however, although similar publications already exist, they have limitations and suggest that further research is needed. This is especially due to the lack of consensus in the understanding of the concept and a lack of acceptance of the concept due to multiple factors, including the scarcity of data to demonstrate the effectiveness of the model or the variability between different health systems around the world.
Comment 4: Pay attention to overlaps with other works. For example: van Staalduinen, D.J., van den Bekerom, P., Groeneveld, S. et al. The implementation of value-based healthcare: a scoping review. BMC Health Serv Res 22, 270 (2022). https://doi.org/10.1186/s12913-022-07489-2
Response: Thank you for your comment. While it is true that our scoping review is similar to the one you refer to, one of the limitations found in that scoping review is the interpretative variability of the concept and the lack of consensus in the conceptualization. These particular aspects have been addressed in our scoping review which has also included relevant studies that have not been taken into account in the scoping review you mentioned.
Reviewer 2 Report
Comments and Suggestions for Authors
Dear authors,
Thank you for your paper. It was interesting to read, however, I think that you can improve the paper with minor changes.
In Chapter 1 I would suggest some further references, especially in the first paragraphs. (e.g. "In this regard, health systems aim to address one of the main concerns about citizen care; the effectiveness of health care outcomes." or "Healthcare organizations are currently facing increased pressure on their total expenditure, increased complexity of people's health status and increased democratization of therapeutic interventions." or " In the traditional medical model, the patient is reduced in his or her relationship with the health system and health professionals to a passive subject and a generic person, i.e., without history or context. This model of health care began to be redefined in the 1970s." All these need a reference to support this statement.
Regarding the provision of care both the costs and the satisfaction, I strongly suggest including other recent references (e.g. https://doi.org/10.3390/ijerph19138188)
Please include some further references regarding the "paradigm of Value-Based Health Care".
The statement "The proposal of the VBHC model responds to the need to address the costs of health services in relation to their capacity to improve the situation of patients." clearly needs some references.
Regarding the GAP identified, it is stated that "...a knowledge gap persists around the existence of consensus on the definition of the VBHC concept." I strongly suggest further explaining how this GAP was identified and back it up with some references.
The paper's objective is clear but can be a bit confusing. "...to identify, compare and summarise the findings of the literature on: 1) The definitions of value-based care extracted from the literature review; 2) the key elements of implementing/delivering value-based care into the healthcare context; and; 3) The main outcomes, in terms of improvement in the quality of the care process, of implementing value-based care. ." I strongly suggest clearing the objective with a more precise scope and a deep explanation to it. Can these 3 points be converted to clusters?
In Chapter 2 I suggest that further references are used to justify the SLR method application. I suggest using reference from "Denyer, D. and Tranfield, D. (2009) Producing a Systematic Review".
Chapter 3. I feel that 12 papers fully reviewed represent a small sample to allow a strong scoping review.
Chapters 4 and 5. The discussion and conclusions are based on the sample articles identified. As a general comment, I think the other Health Services models could have been included in this scoping review. For example, the PPP model is widely accepted and described as VBHC. If the authors agree, I suggest including, at least, in the introduction references to the PPP model (e.g. you can use an SLR already published)
Comments on the Quality of English LanguageI suggest proofreading the document. Some major corrections are necessary.
Author Response
The paper is contributing to the improvement of VBHC concept understanding. In order for various readers to better understanding how the research was conducted, the authors should consider some alterations. Therefore:
Comment 1: In Chapter 1 I would suggest some further references, especially in the first paragraphs. (e.g. "In this regard, health systems aim to address one of the main concerns about citizen care; the effectiveness of health care outcomes." or "Healthcare organizations are currently facing increased pressure on their total expenditure, increased complexity of people's health status and increased democratization of therapeutic interventions." or " In the traditional medical model, the patient is reduced in his or her relationship with the health system and health professionals to a passive subject and a generic person, i.e., without history or context. This model of health care began to be redefined in the 1970s." All these need a reference to support this statement.
Response: We appreciate this suggestion and consequently have included several references in the Introduction to support all ideas.
Comment 2: Regarding the provision of care both the costs and the satisfaction, I strongly suggest including other recent references (e.g. https://doi.org/10.3390/ijerph19138188).
Response: We fully agree with your suggestion and therefore references 18 and 19 have been replaced by more recent references, including specifically the one you have recommended. Thank you.
Comment 3: Please include some further references regarding the "paradigm of Value-Based Health Care".
Response: Thank you very much for your comment. New references have been included to support the Value-Based Health Care paradigm. We appreciate this recommendation.
Comment 4: The statement "The proposal of the VBHC model responds to the need to address the costs of health services in relation to their capacity to improve the situation of patients." clearly needs some references.
Response: Thank you for your comment. New references have been included to support this statement.
Comment 5: Regarding the GAP identified, it is stated that "...a knowledge gap persists around the existence of consensus on the definition of the VBHC concept." I strongly suggest further explaining how this GAP was identified and back it up with some references.
Response: We fully agree with your suggestion, which is why we have explained this statement further in the revised version of the manuscript and included a recent reference to support it.
Comment 6: The paper's objective is clear but can be a bit confusing. "...to identify, compare and summarise the findings of the literature on: 1) The definitions of value-based care extracted from the literature review; 2) the key elements of implementing/delivering value-based care into the healthcare context; and; 3) The main outcomes, in terms of improvement in the quality of the care process, of implementing value-based care. ." I strongly suggest clearing the objective with a more precise scope and a deep explanation to it. Can these 3 points be converted to clusters?
Response: We greatly appreciate your suggestion, but since the aim of the article is threefold and is intended to explore the VBHC model in detail and in depth, it is therefore indicated that all aspects of the model to be explored, i.e. definitions, key elements and results, will be identified, compared and summarised. Although these 3 aspects refer to the VBHC model, they are different and therefore, we find that it is not appropriate to combine or group them together.
Comment 7: In Chapter 2 I suggest that further references are used to justify the SLR method application. I suggest using reference from "Denyer, D. and Tranfield, D. (2009) Producing a Systematic Review".
Response: Thank you very much for your comment and for the example you have set for us. Bearing in mind that this is a Scoping Review, it has finally been decided to include a reference in which more prominence is given to the scoping review and in which the differences and aspects that justify the development of a review as a Systematic review or scoping review are studied.
Comment 8: Chapter 3. I feel that 12 papers fully reviewed represent a small sample to allow a strong scoping review.
Response: Your comment is very much appreciated. We are aware of the small number of articles included in the Scoping Review and therefore this aspect has been highlighted as a limitation. However, after an exhaustive analysis, taking into account the objectives and the established inclusion and exclusion criteria, these are the articles that have fulfilled them.
Comment 9: Chapters 4 and 5. The discussion and conclusions are based on the sample articles identified. As a general comment, I think the other Health Services models could have been included in this scoping review. For example, the PPP model is widely accepted and described as VBHC. If the authors agree, I suggest including, at least, in the introduction references to the PPP model (e.g. you can use an SLR already published)
Response: We appreciate you suggestion of including other Health Services models, but our aim was to focus on the VBHC model. As we want to keep the focus on that, we do not see convenient adding other health care services models to support our ideas.
Comment 10: I suggest proofreading the document. Some major corrections are necessary.
Response: We really appreciate this recommendation and therefore the article has been further revised by a professional linguist to address possible grammatical errors.
Reviewer 3 Report
Comments and Suggestions for Authors
Author Response
Comment 1: At line 142 please include references for MeSH and DeCS.
Response: Thank you for your comment. A new reference has been included to support these concepts.
Comment 2: At line 155: please revise table 2 because the first four lines are identical. Perhaps some descriptors are missing in the last two columns. Table 2 is crucial in the proper understanding of the selection that the authors has made between the 2.801 records they initially considered.
Response: Thank you very much for pointing this out. It was a mistake and those lines that were repeated have been deleted on the revised version of the manuscript.
Comment 3: At line 181: please include a significant reference for PICOTS framework.
Response: Thank you for your comment. A new reference has been included to support the PICOTS framework.
Comment 4: In table 4 please mention as being distinct the two references at Nilsson et al. (2017) as you did in table 3.
Response: Thank you for your appreciation. Both of references have been identified in Table 4 as Nilsson et al. (2017) and Nilsson et al. (2017) b.
Reviewer 4 Report
Comments and Suggestions for Authors
Summary
In this paper, the authors have synthesized and engaged in a discussion about twelve value-based healthcare (VBHC) papers. Firstly, the study delved into the multifaceted definitions of value-based care as distilled from the corpus of existing research. Secondly, it probed the pivotal components indispensable for the effective implementation and delivery of value-based care within the healthcare milieu. Lastly, the investigation synthesized the primary outcomes arising from the integration of value-based care, with a particular focus on its discernible contributions to enhancing the quality of the care process.
Comments:
1) In Table 3, the summaries for both Nilsson (2017)b and Nilsson (2018) appear identical. What distinguishes these two papers from each other?
2) Within Table 4, it is apparent that two entries pertain to Nilsson (2017).
3) In section 3.2.1, the paragraphs lack logical coherence, primarily due to the juxtaposition of concise summaries. The author should explicitly delineate the differing definitions found in the various papers and subsequently expound on the factors contributing to these differences.
4) Within section 3.2.2, it is essential to clarify which specific leadership roles are being referenced and elucidate the interventions undertaken by these leaders. Additionally, it is crucial to specify whether these interventions involve subsidies or regulations.
5) In section 3.2.2, it is imperative to provide a comprehensive explanation of the motivations driving participation in VBHC for patients, healthcare institutions, and hospitals.
6) A detailed analysis is needed in section 3.2.2 to elucidate the role of time in influencing decision changes for patients, healthcare institutions, and hospitals.
7) We strongly recommend that the authors consider enlisting the services of a professional linguist to address grammatical issues and enhance the overall structural coherence of Section 3.2.
Author Response
Comment 1: In Table 3, the summaries for both Nilsson (2017) b and Nilsson (2018) appear identical. What distinguishes these two papers from each other?
Response: Thank you for mentioning this aspect. Both papers refer to exploratory studies exploring the experiences of VBHC implementation, but, while Nilsson (2017)b focuses on exploring how VBHC was instrumental in initiating improvements related to processes, measurements and patient health outcomes, the aim of the study was to gain a deeper understanding of VBHC when used as a management strategy to improve patient health outcomes. While the study of Nilsson (2018) focuses on learning experiences resulting from VBHC implementation. As the papers analyzed different experiences, the results obtained are also different.
Comment 2: Within Table 4, it is apparent that two entries pertain to Nilsson (2017).
Response: Thank you for your appreciation. The Scoping Review includes two articles by Nilsson (2017). They have been identified in Table 4 as Nilsson et al. (2017) a and Nilsson et al. (2017) b.
Comment 3: In section 3.2.1, the paragraphs lack logical coherence, primarily due to the juxtaposition of concise summaries. The author should explicitly delineate the differing definitions found in the various papers and subsequently expound on the factors contributing to these differences.
Response: We really appreciate your suggestion. Table 4 explicitly lists each of the definitions of VBHC for each of the articles included in the scoping review, so to avoid overlap in section 3.2.1 we compare the definitions and explore the similarities and differences between them.
Comment 4: Within section 3.2.2, it is essential to clarify which specific leadership roles are being referenced and elucidate the interventions undertaken by these leaders. Additionally, it is crucial to specify whether these interventions involve subsidies or regulations.
Response: Thank you very much for your appreciation. An additional paragraph has been included mentioning those authors who indicated on which actor the leadership role should fall. The articles included refer to the need for the existence of leadership, and some even consider leadership as an important element as presented in the paper, but do not refer to it as something "normative", which is why it has not been expressed as such.
Comment 5: In section 3.2.2, it is imperative to provide a comprehensive explanation of the motivations driving participation in VBHC for patients, healthcare institutions, and hospitals.
Response: Table 4 shows in detail all the definitions identified in each of the articles included in the scoping review. Therefore, in order to avoid overlapping, section 3.2.1 presents a more concise comparison between the different definitions, analysing the differences and similarities.
Comment 6: A detailed analysis is needed in section 3.2.2 to elucidate the role of time in influencing decision changes for patients, healthcare institutions, and hospitals.
Response: We really appreciate your suggestion of improvement. However, the variable time was not presented in most of the studies and therefore, it cannot be analyzed.
Comment 7: We strongly recommend that the authors consider enlisting the services of a professional linguist to address grammatical issues and enhance the overall structural coherence of Section 3.2.
Response: We really appreciate this recommendation and therefore the article has been further revised by a professional linguist to address possible grammatical errors.